# AI-Based Glioma Grading for a Trustworthy Diagnosis: An Analytical Pipeline for Improved Reliability

**DOI:** 10.3390/cancers15133369

**Published:** 2023-06-27

**Authors:** Carla Pitarch, Vicent Ribas, Alfredo Vellido

**Affiliations:** 1Computer Science Department, Universitat Politècnica de Catalunya (UPC), 08034 Barcelona, Spain; vicent.ribas@eurecat.org (V.R.); avellido@cs.upc.edu (A.V.); 2Eurecat, Technology Centre of Catalonia, Digital Health Unit, 08005 Barcelona, Spain; 3Centro de Investigación Biomédica en Red (CIBER), 28029 Madrid, Spain; 4Intelligent Data Science and Artificial Intelligence Research Center (IDEAI-UPC), 08034 Barcelona, Spain

**Keywords:** glioma, tumor grading, machine learning, decision support, neuro-oncology, radiology, trustworthiness, model certainty, model robustness, reliability

## Abstract

**Simple Summary:**

Accurately grading gliomas, which are the most common and aggressive malignant brain tumors in adults, poses a significant challenge for radiologists. This study explores the application of Deep Learning techniques in assisting tumor grading using Magnetic Resonance Images (MRIs). By analyzing a glioma database sourced from multiple public datasets and comparing different settings, the aim of this study is to develop a robust and reliable grading system. The study demonstrates that by focusing on the tumor region of interest and augmenting the available data, there is a significant improvement in both the accuracy and confidence of tumor grade classifications. While successful in differentiating low-grade gliomas from high-grade gliomas, the accurate classification of grades 2, 3, and 4 remains challenging. The research findings have significant implications for advancing the development of a non-invasive, robust, and trustworthy data-driven system to support clinicians in the diagnosis and therapy planning of glioma patients.

**Abstract:**

Glioma is the most common type of tumor in humans originating in the brain. According to the World Health Organization, gliomas can be graded on a four-stage scale, ranging from the most benign to the most malignant. The grading of these tumors from image information is a far from trivial task for radiologists and one in which they could be assisted by machine-learning-based decision support. However, the machine learning analytical pipeline is also fraught with perils stemming from different sources, such as inadvertent data leakage, adequacy of 2D image sampling, or classifier assessment biases. In this paper, we analyze a glioma database sourced from multiple datasets using a simple classifier, aiming to obtain a reliable tumor grading and, on the way, we provide a few guidelines to ensure such reliability. Our results reveal that by focusing on the tumor region of interest and using data augmentation techniques we significantly enhanced the accuracy and confidence in tumor classifications. Evaluation on an independent test set resulted in an AUC-ROC of 0.932 in the discrimination of low-grade gliomas from high-grade gliomas, and an AUC-ROC of 0.893 in the classification of grades 2, 3, and 4. The study also highlights the importance of providing, beyond generic classification performance, measures of how reliable and trustworthy the model’s output is, thus assessing the model’s certainty and robustness.

## 1. Introduction

Glioma is the most common type of tumor originating in the brain of human adults, and it often comes accompanied by a poor clinical prognosis. In recent decades, the prevalence of brain cancer in the adult population has increased by about 40% [1]. According to the last data available in the Global Cancer Observatory [2], 308,102 people were diagnosed in 2020 as having brain tumors worldwide, with an incidence of 4 per 100,000 people and comprising 1.6% of all new cancer cases. The number of deaths due to brain cancer was 251,329, representing 2.5% of new cancer deaths. These tumors cause different symptoms by pressing on the brain or spinal cord, as they take up space inside the skull when they grow. Headaches, seizures, vision loss, weakness, and speech impairment are among the most common symptoms [3,4].

According to the World Health Organization (WHO) system, gliomas are graded on a scale ranging from 1 (most benign) to 4 (most malignant). In the fourth edition of the WHO Classification of Tumors of the Central Nervous System (CNS) published in 2016, molecular parameters were included along with histological features as decisive markers for glioma classification, mostly based on the presence of isocitrate dehydrogenase (IDH) mutation and codeletion of chromosome arms 1p/19q [5]. The 2021 fifth edition [6] introduced significant changes that advance the role of molecular diagnostics in CNS tumor classification.

Grade 1 gliomas, or pilocytic astrocytomas, are commonly considered different from grades 2–4 and typically occur in children and young adults. They are well-defined, slow-growing, and often curable with complete surgical removal. Gliomas of grade 2 are considered low-grade since they are still slow-growing but are more likely to progress to higher grades. Grade 3 gliomas are more invasive, grow rapidly, and have more abnormal-looking cells. Astrocytomas IDH-mutant and oligodendrogliomas IDH-mutant and 1p/19q codeleted are grade 2–3 tumors that arise, in turn, from astrocytes and oligodendrocytes. Grade 4 gliomas, defined as glioblastomas IDH-wildtype, are the most aggressive and highly invasive. They often contain necrotic areas, which make them difficult to treat, and are associated with poor prognosis.

Treatment options for grades 2, 3, and 4 may include surgery, radiation, and chemotherapy, but the intensity will depend on the aggressiveness of the tumor. Various genetic or molecular features have been identified that play a role in determining the treatment response and prognosis in gliomas, including IDH mutation, 1p/19q codeletion, O-6-methylguanine-DNA methyltransferase (MGMT) methylation, telomerase reverse transcriptase (TERT) promoter mutation, epidermal growth factor receptor (EGFR) amplification, and tumor protein TP53 mutation. Notably, IDH mutation and 1p/19q codeletion are linked to improved prognosis and treatment response. On the other hand, the MGMT promoter has been shown increased sensitivity to alkylating agents such as temozolomide, whereas alterations in the TERT promoter, EGFR amplification, and TP53 mutation are associated with more aggressive phenotype and poorer treatment response [7,8].

The gold-standard procedure for glioma grading involves surgery and histological evaluation of the tumor, which play a pivotal role in both diagnosis and prognosis. However, due to the invasive and time-consuming nature of this method, there is a growing interest in exploring non-invasive and pre-operative procedures to characterize the tumor, accelerate the diagnosis and plan personalized treatments.

The magnetic resonance imaging (MRI) scan is the most widely used test in neurology and neurosurgery. It is a non-invasive method that provides high-level structural and functional information about the brain. Different MRI modalities or sequences are employed for diagnosis, therapy planning, and disease monitoring since they allow the delineation of tumor compartments through visualization of the axial, sagittal, and coronal planes. Conventional MRI modalities include T1, T1-weighted with contrast enhancement, T2-weighted, and FLAIR (fluid-attenuated inversion recovery). Sophisticated MRI systems allow for 3D volume acquisition, which offers high resolution in all three planes. However, the traditional approach consists of acquiring 2D images in a particular anatomical plane and then combining them to create a 3D volume. Commonly, the axial plane is the preferred choice since it allows good visualization of the major structures in the brain.

When glial cells become cancerous, they can have both different appearances and diagnoses. Although glioblastoma is the most common and aggressive brain cancer, it is still a very challenging diagnosis. It is essential to locate the exact brain regions affected by a tumor when planning the treatment and tracking its progression. Automatic reliable identification and characterization of brain tumors for their later removal is a growing public health concern worldwide. The current practice for glioma diagnosis has some associated limitations, such as clinicians’ skills, inter-expert variability, and large waiting times to obtain results. These issues highlight the importance of developing computer-aided diagnosis systems to help radiologists to interpret and quantify abnormalities from brain images through the automation and standardization of tedious and time-consuming tasks involving identification, classification, or segmentation. Algorithms that could at least partially automate the process of tumor localization, monitor the progression and precisely quantify their malignancy, would be very valuable. Radiomics-based approaches have emerged as promising tools in this regard. By extracting quantitative features from medical imaging data, radiomics can contribute not only to the development of a non-invasive and pre-operative system for grading gliomas but also to the characterization of important molecular markers, such as IDH mutation, 1p/19q codeletion, and MGMT promoter methylation, which play crucial roles in treatment decision-making and progression assessment. Numerous studies have previously assessed the classification of the glioma’s grade from MRI using machine learning (ML) methods, but there are still manifold issues in the analytical process that need to be addressed. Furthermore, there is no standardized pipeline that provides guarantees about the methodological issues involved in such analysis. In this study, we strive to gain insight into some of the questions we believe need further investigation. For example,

Do we need to use all the MRI 2D slice images to obtain a reliable classification, or is it enough to use only the slice that contains the largest tumor area?How much additional information is provided by slices that do not contain the largest tumor area?When working with more than one slice for each tumor/patient, a dataset split can be made either at the patient level or slice level. Splitting on slice-level would lead to patient leakage and, if so, can these results be considered trustworthy and generalizable?Most studies highlight overall performance metrics, which, in the very common context of data class imbalance, can prove misleading. Beyond overall classification metrics, which ones would provide scientists trying to replicate experimental settings with the most reliable results?

Beyond answering these questions, this study argues the importance of providing not only classification performance general measures but also measures of how reliable and trustworthy is the model’s output, thus assessing the model’s certainty and robustness.

The main contributions of this work can be summarized as follows:Development of a 2D non-invasive multi-sequence grading system for gliomas.Assessment of the impact of extracting the tumor ROI and using data augmentation techniques on the classification process.A proposal to assess the certainty and robustness of the predictions obtained with the different methods.

### Related Work

Over the last decades, much work has been devoted to the development of at least partially automated systems for glioma grading that could assist clinicians in their diagnostic and prognostic tasks. Recently, most developments concern ML methods and, more in particular, deep learning (DL) techniques. In this section, we briefly review previous works that made a relevant contribution to glioma classification using DL for the analysis of MRI.

Yang et al. [9] proposed the usage of AlexNet and GoogLeNet for classifying LGG and HGG on a private database composed of 113 diagnosed glioma patients. In this study, the 2D T1ce axial images that contain at least 80% of the tumor visible were selected. Then, the image was cropped at the tumor bounding box. The data were split into the train, validation, and test subsets at the patient level. The best performance (AUC = 0.968) was achieved with a pre-trained GoogLeNet. Despite the good overall performance, only accuracy and AUC were reported which, along with the small sample size, are limiting factors in this work.

Pereira et al. [10] proposed a 3D Convolutional Neural Network (CNN) on tumor ROI. By using gradient backpropagation maps, it was found that, by performing *mean-std* intensity standardization in the whole image, the CNN considered the border of the brain as discriminative. Instead, by standardizing the images only with the brain mask, the CNN managed to focus on the tumor, yielding better performance.

Banerjee et al. [11] presented three CNN architectures for classifying the binary grade on MRI patches, slices, and multiplanar 2D MRI by splitting the data at the patient level. They stacked the four conventional MRI sequences as input CNN channels. In this work, a set of 10 slices before and after the slice with the largest tumor area was used, with a skip over five slices for HGG, but with a skip over two slices for LGG, handling the imbalanced dataset problem, but using different criteria for LGG and HGG patients. The best accuracy (0.97) was provided by VolumeNet. Unfortunately, a drawback of this study is that the models were trained on TCGA samples and tested on BraTS2017, which implies an overlap between training and testing data and, therefore, data leakage.

Anaraki et al. [12] proposed the use of a CNN combined with genetic algorithms to select the parameters that lead to the best performance in differentiating healthy images and glioma WHO subtype (2-3-4), by using only T1ce MRI. The reported final accuracy of the selected model was 0.909. The data split in this study seems to be made at the slice level and after data augmentation, which may imply that the same tumor is present multiple times in train and test sets (again, resulting in data leakage). The same practice is conducted by Tandel et al. [13], where authors studied the classification from two categories (tumorous/non-tumorous) up to six different tumor subtypes. The results were reported as the mean of all protocols/models/categories, which makes it difficult to interpret the real class performance. That is, bad performance for some classes can be eclipsed by high performance for other classes. In the classification problem of normal, LGG, or HGG slices, a mean accuracy of 0.960 was obtained, which is the average of K2, K5, and K10 CV protocols.

Zhuge et al. [14] compared two methods for tumor grading (LGG/HGG) by using three MRI modalities (T1-Gd, T2, and Flair). They first used a 3D U-Net to segment the tumor and then compared the grade classification using a 3D volumetric CNN based on NiftyNet [15], and a 2D mask-RCNN on the slice with the largest tumor area. The 2D approach without data augmentation obtained a sensitivity of 0.864, specificity of 0.917, and overall accuracy of 0.891. The 2D approach with data augmentation led to results that were similar to the 3D approach: sensitivity of 0.935, specificity of 0.972, and accuracy of 0.963. The models were trained and validated using Brats2017 and TCGA-LGG data.

Ayadi et al. [16] proposed a model for classifying brain tumor slices in multiple subtypes such as meningioma, glioma, and pituitary tumor and also in glioma sub-grades. Even though the performance of the system proposed in this study is competitive, the training/test partition of the models seems to have been done by image/slice.

Ding et al. [17] proposed a mixed approach using radiomic features from multi-planar reconstructed MRI and DL models, using the slice with the largest tumor area and the two adjacent slices as RGB channels. By combining radiomics and DL, an AUC of 0.898, a sensitivity of 0.840, a specificity of 0.76, and an accuracy of 0.800 were obtained.

van der Voort et al. [18] developed a 3D multitask CNN to predict the grade of the tumor (2/3/4), and mutation status (IDH, 1p/19q co-deletion). Four private datasets and 5 public datasets were merged, including BraTS, EGD, and REMBRANDT. For the grading task, an overall AUC of 0.81, and an accuracy of 0.71 were obtained. A sensitivity of 0.75 for grade 2, 0.17 for grade 3, and 0.95 for grade 4 was reported. When grading in LGG vs. HGG, an AUC of 0.91, an accuracy of 0.84, a sensitivity of 0.72, and a specificity of 0.93 were obtained. Among all the studies reviewed, this paper is the only one providing evidence that distinguishing between grades 2 and 3 is still a very challenging classification problem.

For the sake of clarity, Table 1 summarizes the most relevant results in the reviewed literature that are related to the current study.

## 2. Materials and Methods

### 2.1. Data

For this study, we used datasets from three public repositories, namely BraTS, EGD, and REMBRANDT, which provide pre-operative labeled MR images of gliomas. The brain tumor segmentation (BraTS) challenge [19,20] contains MRI from 2012 to 2021. The challenge aimed to evaluate the state-of-the-art methods for the segmentation of brain tumors, but it also contains clinical information related to the type of tumor. The scans were acquired from 19 institutions and they are already pre-processed and skull-stripped. In this study, we use the BraTS2020 dataset which provides LGG (g.2–g.3) and HGG (g.4) labels for 369 patients. The BraTS database contains samples from The Cancer Genome Atlas, which allowed us to extract the exact grade for the “low-grade” overlapping samples (=65). The Erasmus Glioma Dataset (EGD) [21], provides MR images from 774 patients, and the grade of the tumor for 716 of those. The Cancer Imaging Archive (TCIA) [22] provides raw dicom brain MRI and the tumor grade from 130 patients of the REMBRANDT project [23]. These images were cleaned, transformed to NIfTI format, and segmented by [24]; they are available in the NeuroImaging Tools & Resources Collaboratory (NITRC) resource.

All the scans are co-registered to the same anatomical template (MNI152) and interpolated to a uniform isotropic resolution (1 mm${}^{3}$). All the images are provided in NIfTI format and provide T1, T1ce, T2, and FLAIR modalities, and the segmentation mask of the tumor. After discarding noisy samples, the dataset compiled for the current study contains 1125 samples: 805 HGG and 320 LGG (181 g.2, 122 g.3, 17 unknown). Table A1 provides a comprehensive summary of the patient’s demographic and clinical characteristics in the training (divided into 3 CV folds) and test sets.

#### 2.1.1. Data Pre-Processing

Ranges of image intensities can vary among medical centers, acquisition systems, or clinical protocols. To minimize the effect of these artifacts and solve inhomogeneity issues, applying some pre-processing techniques to MR images [25,26,27] is a common procedure.

One popular step in the MRI pre-processing pipeline is bias field correction (BFC). This method aims to correct the presence of low-frequency intensity non-uniformities in the MRI magnetic field that are commonly known as the bias field. The technique of choice for BFC is N4ITK (N4 bias field correction) [28]. An additional step was required for EGD, since the images are of shape [197, 233, 189], instead of [240, 240, 155] like in BraTS and REMBRANDT. We performed a resample transform using linear interpolation. In both N4ITK and resampling, we used the functions provided by the SimpleITK project.

The images from the BraTS dataset contain only the brain of the patient, so, to homogenize the data, we performed skull-stripping on EGD and REMBRANDT datasets. Skull-stripping is a segmentation task in which brain tissue is segmented from the entire image of the skull [29]. We used the Brain Mask Generator (BrainMaGe) [30] skull-stripping tool, which is built based on a modality-agnostic DL approach. Then, the intensity values of the images were adjusted by redistributing them so that the resulting images match a uniform distribution of intensities. The intensity values are spread out over the full range of possible values, reducing the impact of noise in an image, and enhancing the contrast, in a process that can reveal subtle details.

To unify the intensity levels, the final pre-processing step involved scaling pixel values into the 0–1 range by using min–max normalization. Moreover, each MRI modality image was standardized independently by subtracting the mean and dividing by the standard deviation of the training set and taking only the brain region into account.

The steps performed in the pre-processing pipeline, from raw MRI to the classifier input images, are depicted in Figure 1.

#### 2.1.2. Data Augmentation

Models of the DL family are extremely prone to overfitting the training data when there is insufficient data available. We considered data augmentation techniques to increase both the number of examples and the variability of the training dataset. We artificially augmented the dataset on the fly by making minor alterations to our original data, such as rotations (±90°), random vertical and horizontal flips, random changes in brightness with γ=0.2 and contrast with γ∈(0.25–1.75), and adding random Gaussian noise. By adding the original images to our training set, we ensure that the model learns the key features of the input data and, by including the transformed images, expose the model to a wider range of variations so that it becomes more robust and it generalizes better to unseen data. The augmentation parameter was fine-tuned to determine the best number for generating augmented images, which resulted in two augmented images in addition to the original ones.

Figure 2 shows the experimental pipeline followed in our experiments.

### 2.2. Extraction of the Tumor Region of Interest

We consider two different approaches for glioma grading: either involving the whole brain or extracting the tumor region of interest (ROI). We defined a bounding box around the tumor area from the tumor segmentation mask to extract the tumor ROI. After extracting the bounding box, the image was resized to a fixed size of 128×128 to be fed into the classifier.

### 2.3. Classification Model

Over the last few years, the use of DL to automatically analyze images has revolutionized the field of computer vision. The current gold-standard DL architecture in computer vision is the convolutional neural network (CNN) [31]. These models can be fed by one, two, or three-dimensional inputs. The input of a CNN is a tensor with a shape *(width × height × channels)*. In our particular case, we modified the input number of channels of the network to four, each representing an MRI modality.

Our baseline models were developed based on the selection of the single 2D MRI slice that contained the largest area of the tumor. This selection was carried out by maximizing the inclusion of tumor pixels as indicated by the tumor mask. Additionally, we explored an alternative approach by incorporating 20 consecutive slices where the tumor was visibly present. These additional slices encompassed the 10 preceding and 10 succeeding slices originating from the 2D slice with the maximum tumor area. The final patient diagnosis was achieved by employing a majority voting approach.

Our models were built using the ResNet18 [32] architecture, consisting of 18 layers, including convolutional, pooling, fully connected layers, and residual blocks. These residual connections allow the network to propagate gradients effectively, reducing the vanishing gradient problem that can occur in very deep networks. In addition to the original architecture, we added a dropout layer before the last fully connected layer to prevent overfitting. Finally, a Softmax layer was added to convert logits into probabilities. We chose ResNet18 among different models, such as AlexNet [33], VGGNet11 [34], VGGNet16, or ResNet34 since they lead us to worse performance or overfitting when the networks were deeper.

In our experiments, the cross-entropy loss function was used as the objective function for training the deep neural network models. However, due to the imbalance problem in the analyzed datasets, a weighted cross-entropy loss function was implemented to mitigate this effect. This approach involved assigning weights to each class to give greater importance to the underrepresented classes during training.

To optimize the model, the SGD optimizer [35] was used, with a learning rate of 1 × 10${}^{-4}$, a momentum of 0.9, and a weight decay of 5 × 10${}^{-4}$. A batch size of 32 and a maximum number of epochs of 100 were set while saving the model at the epoch that achieved the best performance in the validation set. Finally, the network was initialized using Kaiming weights initialization [36], which also helps avoid vanishing gradients. These hyperparameters were selected based on a grid search over a range of values and were found to produce good results on our dataset.

A 3-fold cross-validation (CV) was used to assess the effectiveness of our models. A total of 75% of the data was used to run the 3-fold CV while keeping 25% as a holdout test set. Class proportions were preserved in each set. In comparison to using a single train/test split, this enables us to get more reliable estimations of the model’s performance. The final model’s performance is presented by averaging the three folds. After completing the CV, we retrained the model using a random split of the train, validation, and test sets. This was done to ensure that the performance of the model was not biased by the specific partitioning used during cross-validation.

In order to enhance the robustness and reliability of our findings, an ensemble approach was implemented. This involved gathering the test predictions from each fold of the CV process and integrating them by averaging the probability outputs of the models. This resulted in a final prediction for each sample in the test set. This ensemble approach allows us to combine the predictions of multiple models, each trained on a different subset of the data, to obtain a more robust prediction that is less likely to be influenced by noise or outliers. Ensembling can help reduce the impact of any individual fold’s weaknesses or biases. We believe that this approach provides a more reliable estimate of the model’s performance and increases confidence in our results.

The whole process was conducted using an NVIDIA GRID A100-20C GPU with CUDA version 12.0 and the models were implemented using PyTorch 1.11.0.

### 2.4. Performance Metrics

The confusion matrix and its associated performance metrics were used to evaluate the predictive performance of our models. These metrics provide information about the classification algorithms’ accuracy, precision, recall, and overall performance, and will allow us to assess their ability to correctly classify instances within our dataset. The confusion matrix (Table 2) was used as the basis to evaluate the predictive performance of our models, where *TP* stands for true positive, *TN* for true negative, *FP* for false positive, and *FN* for false negative.

Several standard metrics can be obtained from the confusion matrix, including $accuracy=\frac{TP+TN}{TP+FP+TN+FN}$, $sensitivity=\frac{TP}{TP+FN}$, $specificity=\frac{TN}{TN+FP}$, $Precision=\frac{TP}{TP+FP}$, and the $F1Score=2\frac{Recall*Precision}{Recall+Precision}$.

Accuracy may be not suitable when working in imbalanced domains because by always correctly predicting the most represented class (even when no observations of the minority class are correctly classified) a very high but misleading accuracy can still be obtained.

A far more robust-to-class-imbalance metric is the *Area Under the Roc Curve (AUC)*, which can be interpreted as the probability that a random individual who will become positive has a higher risk of being positive than a negative individual. The AUC integrates measures of the discriminative ability of an algorithm across different thresholds, and can be computed as follows:(1)$$AUC_{ROC}=\int_{0}^{1}\frac{TP}{TP+FN}d\left(\frac{FP}{TN+FP}\right)$$

An AUC close to 1 means the model is able to separate the classes perfectly, whereas an AUC value close to 0.5 is the signature of a completely random classification model.

As the class imbalance problem can significantly impact the metrics derived from the confusion matrix, we provide them for each class.

## 3. Results

Given that the main objective of this work is providing guidelines about the different elements in the pipeline of ML-based image analysis that are required to enhance the reliability of such analysis, we initially conducted a comprehensive comparison of performance and certainty metrics across various settings, which included either using tumor patches or considering the entire brain region, as well as either incorporating data augmentation techniques or using unmodified data. Specifically, we employed the slice with the largest tumor area as a baseline approach. Subsequently, we trained a model using the settings that yielded the best results by incorporating multiple slices, aiming to gain insights into the additional benefit contributed by this approach to the classification process.

### 3.1. *Grade Classification Performance*

Table 3 and Table 4 report the classification performance measures obtained without using data augmentation techniques in the skull-stripped brain images and tumor ROIs, respectively. Similarly, Table 5 and Table 6 provide the analogous results obtained using data augmentation transforms. Each table contains the results for both binary (LGG vs HGG) and multi-class (g.2, g.3, and g.4) scenarios.

We observed a significant improvement in the classification metrics for classifying the grades in LGG (grades 2 and 3) and HGG (grade 4). The ensemble results obtained from the independent test set show an improvement in the sensitivity of LGG and HGG from 0.725 and 0.748 to 0.863 and 0.891, respectively. The accuracy and the AUC-ROC are boosted from 0.741 and 0.819 to 0.883 and 0.932.

The findings from the multi-class problem are revealing in several ways. Accuracy for each grade improved from 0.835, 0.892, 0.734 to, in turn, 0.874, 0.892, and 0.881, leading to an overall accuracy of 0.730 and an AUC of 0.671. Further, the AUC-ROC for each category increased from 0.705, 0.636, and 0.671 to, in turn, 0.921, 0.825, and 0.935, which resulted in an overall accuracy of 0.824 and AUC of 0.893. A closer inspection of the results shows that the sensitivity for grades 2, 3, and 4 raised from 0.022, 0.000, and 1.000 to 0.870, 0.100, and 0.916. Despite the high accuracy and AUC, these results suggest that we were able to accurately classify 87% of grade 2 and 91.6% of grade 4 tumors, but only 10% of grade 3 tumors.

From Figure 3, it can be seen that the use of tumor ROIs and data augmentation techniques enhances both the model’s classification performance and its generalization ability. Additionally, the curves appear to be less noisy, indicating that the model’s predictions are more consistent and reliable.

To evaluate the similarity of data distributions across the three folds and test sets, statistical tests were conducted to examine the presence of any significant differences in grade and tumor size. The findings of these tests, as shown in Table A2 and Table A3, indicate that no significant differences exist among the various datasets.

### 3.2. *Single-Slice versus Multi-Slice*

The outcomes obtained from the inclusion of multiple consecutive slices are illustrated in Table 7. The recall for LGG and HGG classification resulted in 0.887 and 0.881, respectively. In the case of WHO grade classification, the proportion of correctly classified observations for grade 2 was 0.913, for grade 3 was 0.100, and for grade 4 was 0.926. The overall accuracy and AUC values also remained consistent between the two approaches.

These results did not exhibit substantial improvement compared to the single-slice approach, indicating that the additional information provided by multiple slices did not significantly contribute to the classification task.

### 3.3. *Model Robustness and Certainty*

Figure 4 and Figure 5 present histograms that illustrate the distribution of model output probabilities in the test set for each of the three trained models in the respective classification tasks. In turn, Figure 6 and Figure 7 offer a different perspective by quantifying the certainty of the model’s classifications. These figures distinguish between correct and incorrect classifications and depict the level of certainty in the predictions.

In the context of binary classification, in Figure 4D, we can observe a concentration of probabilities around 1 when using tumor ROIs and data augmentation. Similarly, Figure 6D illustrates the largest number of accurate and confident classifications. These outcomes indicate a strong level of confidence in the model’s predictions.

The plots depicting the WHO grade classification scenario reaffirm our earlier conclusion that the model struggled to differentiate grade 3 from grades 2 and 4. Nevertheless, the model demonstrated a moderate level of confidence in predicting grade 2 tumors, while yielding a high level of confidence in predicting grade 4 tumors. Once more, Figure 5D and Figure 7D emphasize that the most reliable classification is achieved by employing tumor ROIs and data augmentation.

In summary, our results provide compelling evidence that integrating tumor ROIs and employing data augmentation techniques significantly enhances the accuracy, confidence, and robustness of our predictions.

## 4. Discussion

The primary objective of this study was to develop a reliable and transparent DL-based method for glioma grading. To achieve this, we conducted a comparative analysis to assess the certainty and robustness of the predictions generated by employing various strategies, including data augmentation and focusing on the tumor ROI. By addressing the need for a non-invasive, accurate, and trustworthy grading system, this study aims to contribute to the field of neuro-oncology.

Several prior studies have also focused on developing DL-based systems for glioma grading. One common challenge in these studies is handling data imbalance and small sample sizes. To address this issue, three popular public datasets such as BraTS, TCGA, and Rembrandt are commonly used for benchmarking ML glioma classification systems. In our study, we also incorporated the EGD, aiming to enhance the generalizability and robustness of our findings. The inclusion of multiple datasets with varying imaging protocols, patient populations, and tumor characteristics allowed us to validate the consistency of our results.

The pre-processing pipeline is shown to play a crucial role in achieving dataset harmonization. Initially, we implemented commonly employed techniques to correct brain MRI artifacts, which encompassed resampling the images to a common shape, performing skull stripping to remove non-brain tissue, and applying bias field correction to compensate for intensity variations. Furthermore, before inputting the images into the model, we normalized and standardized the image intensities using the mean and standard deviation extracted from brain pixels.

By combining the predictions obtained from the independent evaluation of the test set using a threefold CV approach, we accounted for the inherent variability in our data and mitigated any potential bias introduced during the data-splitting process. Notably, the model trained with data augmentation transforms and focused on tumor ROIs yielded more accurate and robust predictions in the unseen test set. Our approach achieved an overall AUC of 0.932 for distinguishing between LGG (g.2–g.3) and HGG (g.4 or glioblastomas), as well as an AUC of 0.893 for the WHO grade classification (g.2, g.3, and g.4). When our model demonstrated a sensitivity of 0.864 and 0.891 for classifying LGG and HGG, respectively, it faced challenges in accurately discriminating the WHO grade. Notably, the sensitivity for grade 3 was only 0.100, whereas for grades 2 and 4, it was 0.87 and 0.921, respectively. The results align with the study conducted by van der Voort et al. (2023) [18], who reported comparable classification metrics regarding the distinction between WHO grades. These findings support the conclusion that classifying grade 3 tumors using MRI data remains a challenge for which images might not be sufficient. The use of MR spectroscopy (MRS) data, or even multimodal data such as MRSI, might enhance the grading classification results.

3D MRIs can be decomposed into individual 2D slices, each providing a distinct perspective of the tumor with a typical spacing of around 1 mm${}^{3}$ between the slices. To ensure the validity of the classification performance, it is crucial to perform the data split at the patient level rather than the individual image level when working with multiple slices per patient. This approach minimizes the risk of including 2D slices from the same patient in both the training and testing sets. This prevents a potential over-optimistic estimation of the classification performance and ensures the independence and integrity of the training, validation, and testing sets. Although studies presented by Anaraki et al. (2019) [12], Tandel et al. (2020) [13], and Ayadi et al. (2021) [16] achieved competitive performance in brain tumors classification, it is worth noting that they performed the split at the slice-level, therefore not addressing the potential problem of data leakage.

This analysis has led us to a significant conclusion, indicating that working solely with 2D slices may not capture the complete 3D context and interrelationships among slices originating from the same 3D image. In a 2D analysis, each slice is treated independently disregarding the spatial information and correlations with adjacent slices. This approach can limit the ability to capture the entire tumor structure and the contextual information present in the 3D volume. Consequently, the incorporation of multiple slices may not yield substantial additional information beyond what is already captured by the slice with the largest tumor area. These findings emphasize the significance of considering the 3D context in future research endeavors and exploring alternative approaches that leverage the spatial relationship between slices.

## 5. Conclusions

In this study, we have developed a non-invasive DL-based analytical pipeline using together the four conventional MRI modalities for classifying glioma grades. The pipeline is meant to ensure the robustness and reliability of our predictions. Although we achieved promising results in characterizing grades 2 and 4, further research is needed to develop a reliable system for distinguishing between the four glioma grades. Importantly, though, we have complemented the more standard classification performance results with accompanying quantification of the model certainty on its predictions.

Although merging different public datasets helped address the issue of limited sample size, additional efforts are needed to acquire large harmonized databases in this medical domain, which would, by themselves, enhance the reliability of the DL-based pipeline.

There is an opportunity to further enhance glioma grading in future research by employing a 3D approach that takes into account spatial context information. The significance of molecular biomarkers in the brain tumor diagnostic process has been underscored by the WHO glioma categorization [6]. By merging clinical data and imaging data, there is also an opportunity to achieve improved differentiation between low-grade gliomas. This integration of multi-dimensional information holds the promise of enhancing the accuracy and reliability of glioma grading systems, ultimately leading to more precise diagnoses and more effective treatment planning in clinical practice.

## Figures and Tables

**Figure 1 cancers-15-03369-f001:**
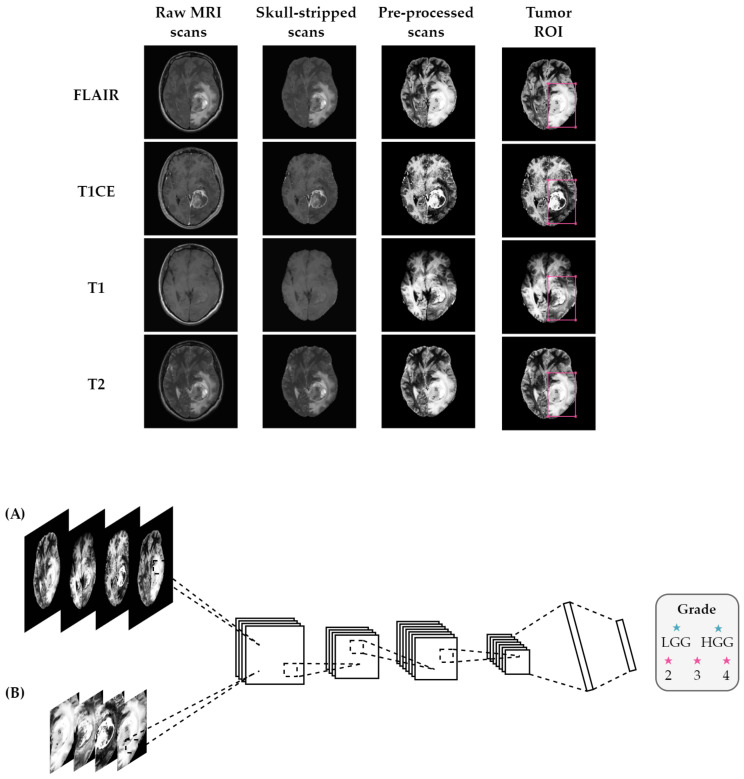
Overview of the proposed method using (**A**) entire brain images and (**B**) tumor ROI. Before feeding the MRI scans into the classifier, the pipeline consisted of several pre-processing steps, including registration to a common atlas, skull-stripping, bias field correction, and normalization. The classifier takes FLAIR, T1 with contrast-enhancement, T1, and T2 scans stacked as input channels for the classification task.

**Figure 2 cancers-15-03369-f002:**
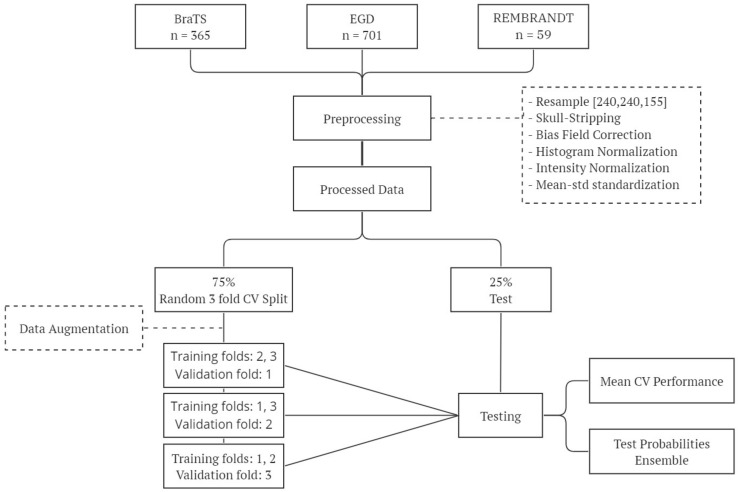
Overview of the experimental workflow: this diagram outlines the key steps involved in our experimental methodology, including data preparation, training, and testing.

**Figure 3 cancers-15-03369-f003:**
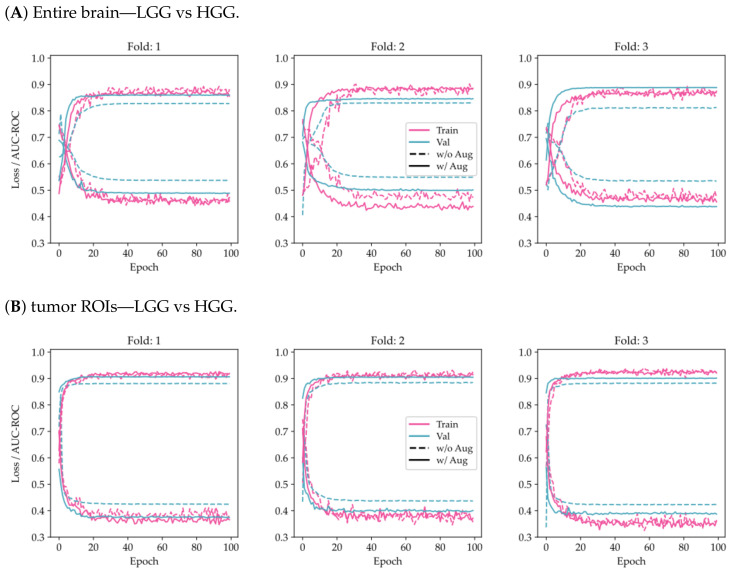
Comparison of training and validation loss and AUC-ROC history using the proposed model on a 3-fold CV for the two-class problem (LGG/HGG) and the multi-class problem (g.2/g.3/g.4). The results are shown for two different scenarios: (**A**,**C**) considering the entire brain, and (**B**,**D**) considering tumor ROIs.

**Figure 4 cancers-15-03369-f004:**
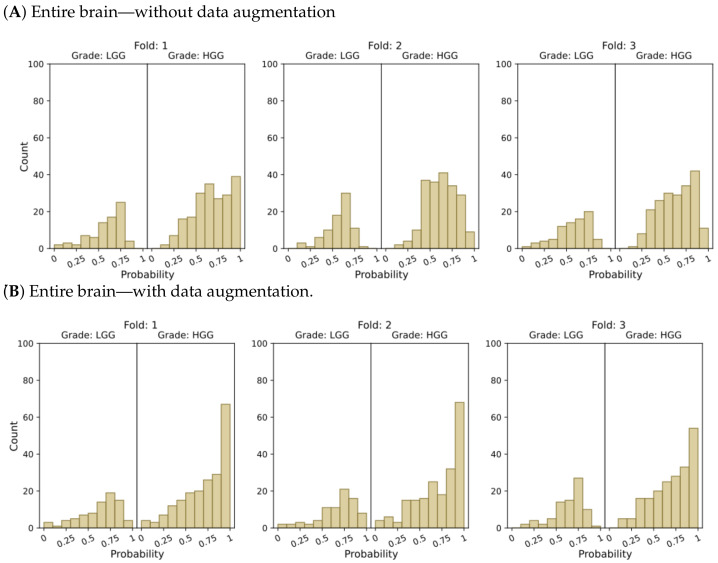
Probability distributions of model predictions for LGG and HGG classification, using (**A**) the entire brain without data augmentation, (**B**) the entire brain with data augmentation, (**C**) tumor ROIs without data augmentation, and (**D**) tumor ROIs with data augmentation.

**Figure 5 cancers-15-03369-f005:**
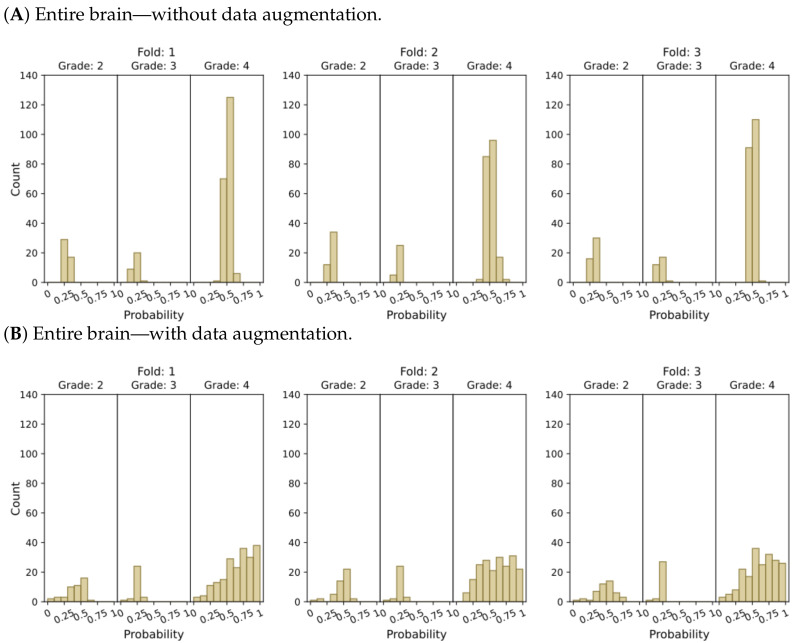
Model’s output probability distributions for WHO glioma grade classification (g.2, g.3, g.4), using (**A**) the entire brain without data augmentation, (**B**) the entire brain with data augmentation, (**C**) tumor ROIs without data augmentation, and (**D**) tumor ROIs with data augmentation.

**Figure 6 cancers-15-03369-f006:**
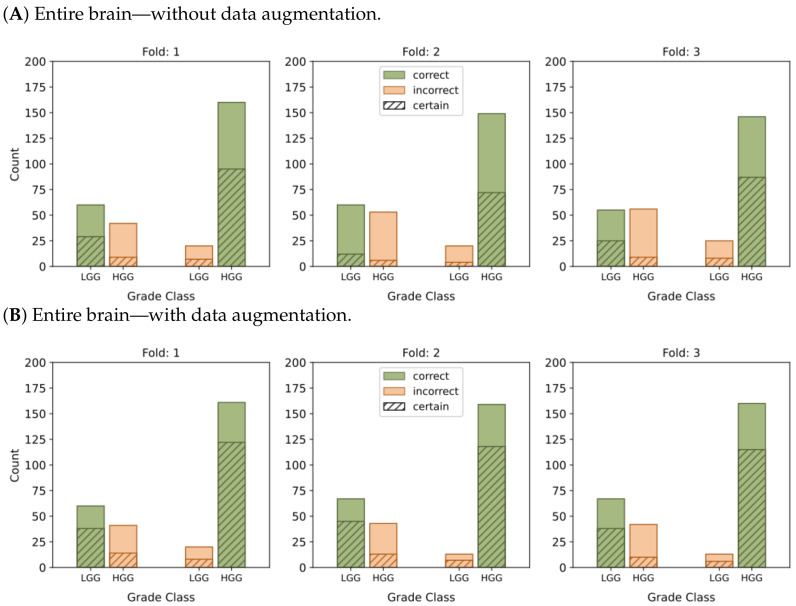
This graphic represents the categorization of model predictions into certain (probability ≥ 0.7) and uncertain (probability < 0.7), as well as the accuracy of each of the four models, namely (**A**–**D**), in classifying LGG and HGG samples. Correct classifications are shown in green while incorrect classifications are shown in orange.

**Figure 7 cancers-15-03369-f007:**
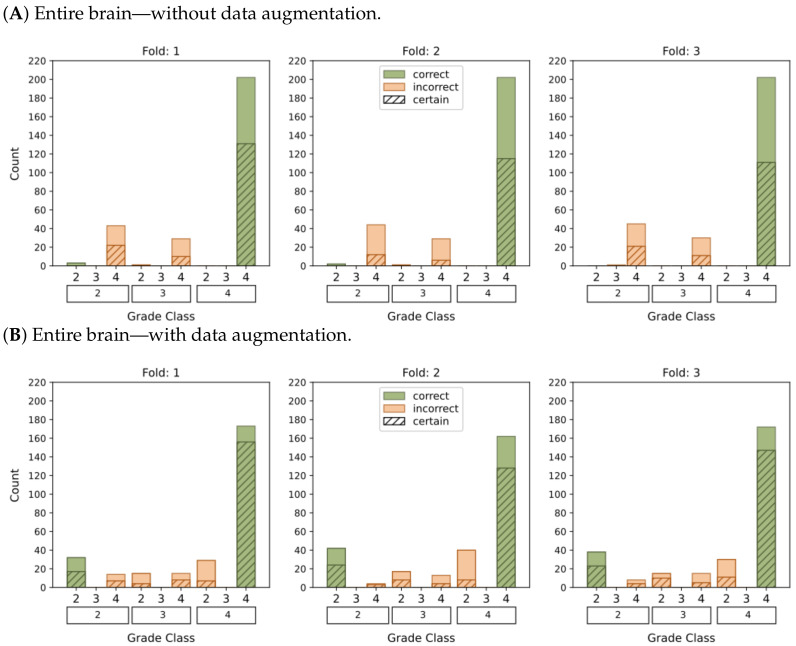
This graphic represents the categorization of model predictions into certain (probability ≥ 0.5) and uncertain (probability < 0.5), as well as the accuracy of each of the four models (**A**–**D**) in classifying glioma grades in WHO categories (g.2, g.3, g.4). Correct classifications are shown in green whereas incorrect classifications are shown in orange.

**Table 1 cancers-15-03369-t001:** Summary of relevant literature on glioma grading and the corresponding methodologies used. For each study, we report details about the database used, sample size, MRI modality, data spitting strategy, dimensionality, number of slices used, categories considered, and the reported performance.

Reference	Database	Sample Size	MRI Modality	Split	Dimensionality	Slices	Classes	Performance
Yang et al. [9]	Private	113	T1ce	Patient	2D	≥80% tumor	LGG/HGG	OA = 0.945
								AUC = 0.968
Pereira et al. [10]	BraTS2017	285	T1, T1ce, T2, FLAIR	Patient-level	3D	All	LGG/HGG	OA = 0.9298
								AUC = 0.9841
								Recall LGG = 0.867
								Recall HGG = 0.952
Banerjee et al. [11]	BraTS2017	746	T1, T1ce, T2, FLAIR	Patient	2D	largest slice ± 10	LGG/HGG	OA = 0.970
	TCGA							Recall LGG = 0.96
								Recall HGG = 0.943
Anaraki et al. [12]	IXI	1288	T1ce	Slice	2D	tumor visible	Normal/G.2/G.3/G.4	OA = 0.909
	REMBRANDT							Recall Normal = 0.998
	TCGA							Recall G.2 = 0.884
	private							Recall G.3 = 0.864
								Recall G.4 = 0.974
Tandel et al. [13]	REMBRANDT	130	T2	Slice	2D	tumor visible	Normal/LGG/HGG	OA = 0.960
								AUC = 0.990
Zhuge et al. [14]	Brats2018	315	T1ce, T2, FLAIR	Patient	2D	largest slice	LGG/HGG	OA = 0.963
	TCGA							Recall HGG = 0.935
								Recall LGG = 0.972
				Patient	3D	All	LGG/HGG	OA = 0.971
								Recall HGG = 0.947
								Recall LGG = 0.968
Ayadi et al. [16]	REMBRANDT	130		Slice	2D	tumor visible	Normal/LGG/HGG	OA = 0.972
								Recall Normal = 1
								Recall LGG = 0.984
								Recall HGG = 0.860
	Radiopaedia	121		Slice	2D	tumor visible	G.1/G.2/G.3/G.4	OA = 0.937
								Recall G.1 = 0.901
								Recall G.2 = 0.957
								Recall G.3 = 0.908
								Recall G.4 = 0.982
Ding et al. [17]	Private	151	T1ce	Patient	2D	largest + 2 adjacent	LGG/HGG	OA= 0.800
	TCGA							AUC = 0.898
								Recall LGG = 0.760
								Recall HGG = 0.840
van der Voort et al. [18]	EGD	1748	T1, T1ce, T2, FLAIR	Patient	3D	All	LGG/HGG	OA = 0.84
	REMBRANDT							AUC = 0.91
	CPTAC-GBM							Recall LGG = 0.72
	TCGA							Recall HGG = 0.93
	BraTS2019						G.2/G.3/G.4	OA = 0.71
	Ivy GAP							AUC = 0.81
	Private							Recall G.2 = 0.75
								Recall G.3 = 0.17
								Recall G.4 = 0.95

OA = Overall Accuracy, AUC = Area Under the (ROC) Curve.

**Table 2 cancers-15-03369-t002:** Confusion matrix.

		Predicted	
		Negative	Positive
True	Negative	TN	FP
	Positive	FN	TP

**Table 3 cancers-15-03369-t003:** Performance evaluation of the model trained on the largest slice using entire brain images without data augmentation.

	Mean 3-Fold CV			Ensemble 3-Fold CV
	**Train**	**Validation**	**Test**	**Test** **(95% CI)**
**LGG vs. HGG**				
Loss	0.479	0.540	0.546	
Accuracy	0.780	0.769	0.745	0.741 (0.690–0.792)
AUC-ROC	0.872	0.823	0.807	0.819 (0.774–0.864)
Precision				
LGG	0.582	0.572	0.538	0.532 (0.474–0.590)
HGG	0.914	0.884	0.875	0.873 (0.834–0.912)
Recall				
LGG	0.821	0.742	0.729	0.725 (0.673–0.777)
HGG	0.764	0.779	0.751	0.748 (0.697–0.798)
F1				
LGG	0.681	0.646	0.643	0.614 (0.557–0.671)
HGG	0.832	0.828	0.824	0.805 (0.759–0.852)
**Grade (2/3/4)**				
Loss	1.009	1.006	0.987	
Accuracy	0.647	0.725	0.733	0.730 (0.678–0.782)
G.2	0.749	0.837	0.838	0.835 (0.791–0.878)
G.3	0.860	0.886	0.891	0.892 (0.856–0.929)
G.4	0.684	0.728	0.736	0.734 (0.682–0.786)
AUC-ROC	0.646	0.642	0.651	0.671 (0.616–0.726)
G.2	0.639	0.684	0.667	0.705 (0.652–0.759)
G.3	0.642	0.595	0.635	0.636 (0.580–0.693)
G.4	0.658	0.647	0.650	0.671 (0.616–0.726)
Precision				
G.2	0.275	0.167	0.472	0.500 (0.441–0.559)
G.3	0.226	0.000	0.000	0.000 (0.000–0.000)
G.4	0.773	0.729	0.734	0.732 (0.680–0.784)
Recall				
G.2	0.318	0.008	0.036	0.022 (0.005–0.039)
G.3	0.114	0.000	0.000	0.000 (0.000–0.000)
G.4	0.802	0.997	1.000	1.000 (1.000–1.000)
Specificity				
G.2	0.833	0.999	0.997	0.996 (0.988–1.000)
G.3	0.953	0.996	0.999	1.000 (1.000–1.000)
G.4	0.372	0.013	0.035	0.026 (0.007–0.045)
F1				
G.2	0.293	0.014	0.028	0.042 (0.018–0.065)
G.3	0.152	0.000	0.000	0.000 (0.000–0.000)
G.4	0.787	0.842	0.842	0.845 (0.803–0.888)

**Table 4 cancers-15-03369-t004:** Performance evaluation of the model trained on the largest slice using tumor ROI without data augmentation.

	Mean 3-Fold CV			Ensemble 3-Fold CV
	**Train**	**Validation**	**Test**	**Test** **(95% CI)**
**LGG vs. HGG**				
Loss	0.379	0.427	0.386	
Accuracy	0.854	0.821	0.829	0.837 (0.794–0.880)
AUC-ROC	0.913	0.883	0.906	0.910 (0.877–0.944)
Precision				
LGG	0.703	0.656	0.665	0.677 (0.623–0.732)
HGG	0.932	0.905	0.916	0.919 (0.888–0.951)
Recall				
LGG	0.842	0.779	0.804	0.812 (0.767–0.858)
HGG	0.858	0.838	0.838	0.847 (0.804–0.889)
F1				
LGG	0.766	0.712	0.714	0.739 (0.687–0.790)
HGG	0.893	0.870	0.872	0.881 (0.844–0.919)
**Grade (2/3/4)**				
Loss	0.690	0.762	0.685	
Accuracy	0.807	0.788	0.797	0.799 (0.751–0.846)
G.2	0.877	0.840	0.853	0.856 (0.815–0.897)
G.3	0.881	0.889	0.891	0.892 (0.856–0.929)
G.4	0.856	0.847	0.852	0.849 (0.807–0.891)
AUC-ROC	0.871	0.813	0.854	0.860 (0.819–0.901)
G.2	0.924	0.890	0.909	0.911 (0.878–0.945)
G.3	0.787	0.669	0.752	0.765 (0.715–0.815)
G.4	0.901	0.880	0.902	0.904 (0.870–0.939)
Precision				
G.2	0.592	0.507	0.535	0.543 (0.484–0.601)
G.3	0.446	0.000	0.000	0.000 (0.000–0.000)
G.4	0.912	0.884	0.890	0.885 (0.847–0.922)
Recall				
G.2	0.807	0.785	0.833	0.826 (0.782–0.871)
G.3	0.281	0.000	0.000	0.000 (0.000–0.000)
G.4	0.887	0.909	0.908	0.911 (0.877–0.944)
Specificity				
G.2	0.891	0.850	0.856	0.862 (0.822–0.903)
G.3	0.956	1.000	0.999	1.000 (1.000–1.000)
G.4	0.773	0.683	0.702	0.684 (0.630–0.739)
F1				
G.2	0.681	0.616	0.617	0.655 (0.599–0.711)
G.3	0.344	0.000	0.000	0.000 (0.000–0.000)
G.4	0.899	0.896	0.894	0.898 (0.862–0.933)

**Table 5 cancers-15-03369-t005:** Performance evaluation of the model trained on the largest slice using entire brain images and data augmentation.

	Mean 3-Fold CV			Ensemble 3-Fold CV
	**Train**	**Validation**	**Test**	**Test** **(95% CI)**
**LGG vs. HGG**				
Loss	0.451	0.474	0.459	
Accuracy	0.793	0.797	0.797	0.805 (0.759–0.851)
AUC-ROC	0.874	0.864	0.866	0.878 (0.840–0.916)
Precision				
LGG	0.600	0.611	0.606	0.624 (0.567–0.680)
HGG	0.917	0.911	0.913	0.906 (0.872–0.940)
Recall				
LGG	0.822	0.804	0.809	0.788 (0.740–0.835)
HGG	0.781	0.794	0.792	0.812 (0.766–0.857)
F1				
LGG	0.693	0.694	0.701	0.696 (0.642–0.750)
HGG	0.843	0.848	0.856	0.856 (0.815–0.897)
**Grade (2/3/4)**				
Loss	0.867	0.837	0.802	
Accuracy	0.729	0.751	0.742	0.752 (0.701–0.803)
G.2	0.801	0.793	0.794	0.802 (0.755–0.849)
G.3	0.865	0.889	0.892	0.892 (0.856–0.929)
G.4	0.793	0.819	0.798	0.809 (0.763–0.856)
AUC-ROC	0.776	0.788	0.803	0.816 (0.770–0.861)
G.2	0.838	0.849	0.862	0.872 (0.833–0.911)
G.3	0.865	0.670	0.697	0.713 (0.660–0.766)
G.4	0.793	0.844	0.850	0.861 (0.821–0.902)
Precision				
G.2	0.427	0.422	0.434	0.448 (0.390–0.507)
G.3	0.246	0.000	0.000	0.000 (0.000–0.000)
G.4	0.869	0.880	0.881	0.890 (0.853–0.927)
Recall				
G.2	0.655	0.726	0.812	0.848 (0.806–0.890)
G.3	0.099	0.000	0.000	0.000 (0.000–0.000)
G.4	0.842	0.871	0.836	0.842 (0.799–0.885)
Specificity				
G.2	0.829	0.806	0.790	0.793 (0.745–0.841)
G.3	0.960	1.000	1.000	1.000 (1.000–1.000)
G.4	0.662	0.683	0.697	0.724 (0.671–0.776)
F1				
G.2	0.517	0.532	0.520	0.586 (0.529–0.644)
G.3	0.139	0.000	0.000	0.000 (0.000–0.000)
G.4	0.662	0.875	0.872	0.865 (0.825–0.905)

**Table 6 cancers-15-03369-t006:** Performance evaluation of the model trained on the largest slice using tumor ROI and data augmentation.

	Mean 3-Fold CV			Ensemble 3-Fold CV
	**Train**	**Validation**	**Test**	**Test** **(95% CI)**
**LGG vs. HGG**				
Loss	0.375	0.384	0.335	
Accuracy	0.841	0.844	0.870	0.883 (0.845–0.920)
AUC-ROC	0.913	0.905	0.928	0.932 (0.902–0.961)
Precision				
LGG	0.680	0.685	0.732	0.758 (0.708–0.808)
HGG	0.927	0.931	0.940	0.942 (0.915–0.970)
Recall				
LGG	0.834	0.842	0.858	0.863 (0.822–0.903)
HGG	0.844	0.846	0.874	0.891 (0.855–0.927)
F1				
LGG	0.749	0.755	0.759	0.807 (0.761–0.853)
HGG	0.883	0.886	0.890	0.916 (0.884–0.948)
**Grade (2/3/4)**				
Loss	0.722	0.712	0.617	
Accuracy	0.780	0.798	0.818	0.824 (0.779–0.869)
G.2	0.847	0.854	0.869	0.874 (0.835–0.913)
G.3	0.852	0.883	0.885	0.892 (0.856–0.929)
G.4	0.847	0.858	0.881	0.881 (0.843–0.919)
AUC-ROC	0.847	0.837	0.886	0.893 (0.857–0.930)
G.2	0.915	0.909	0.920	0.921 (0.889–0.952)
G.3	0.724	0.701	0.809	0.825 (0.780–0.869)
G.4	0.903	0.901	0.929	0.935 (0.780–0.869)
Precision				
G.2	0.554	0.534	0.570	0.580 (0.522–0.638)
G.3	0.241	0.400	0.381	0.500 (0.441–0.559)
G.4	0.911	0.905	0.920	0.916 (0.884–0.949)
Recall				
G.2	0.781	0.837	0.855	0.870 (0.830–0.909)
G.3	0.154	0.076	0.100	0.100 (0.065–0.135)
G.4	0.875	0.899	0.916	0.921 (0.889–0.953)
Specificity				
G.2	0.877	0.858	0.872	0.875 (0.836–0.914)
G.3	0.939	0.984	0.980	0.988 (0.975–1.000)
G.4	0.772	0.749	0.789	0.776 (0.727–0.825)
F1				
G.2	0.647	0.651	0.663	0.696 (0.642–0.750)
G.3	0.187	0.125	0.122	0.167 (0.123–0.210)
G.4	0.893	0.902	0.907	0.919 (0.886–0.951)

**Table 7 cancers-15-03369-t007:** Performance evaluation of the model trained on 20 consecutive slices using tumor ROI with data augmentation.

	Mean 3-Fold CV			Ensemble 3-Fold CV
	**Train**	**Validation**	**Test**	**Test** **(95% CI)**
**LGG vs. HGG**				
Loss	0.354	0.336	0.371	
Accuracy	0.894	0.883	0.882	0.883 (0.845–0.920)
AUC-ROC	0.923	0.922	0.921	0.927 (0.896–0.957)
Precision				
LGG	0.772	0.767	0.752	0.747 (0.697–0.798)
HGG	0.952	0.935	0.945	0.952 (0.927–0.977)
Recall				
LGG	0.888	0.843	0.871	0.887 (0.851–0.924)
HGG	0.896	0.898	0.886	0.881 (0.843–0.919)
F1				
LGG	0.825	0.804	0.807	0.811 (0.766–0.857)
HGG	0.923	0.916	0.915	0.915 (0.883–0.948)
**Grade (2/3/4)**				
Loss	0.696	0.590	0.603	
Accuracy	0.839	0.818	0.810	0.835 (0.791–0.878)
G.2	0.895	0.870	0.863	0.878 (0.839–0.916)
G.3	0.896	0.877	0.872	0.896 (0.860–0.932)
G.4	0.889	0.888	0.884	0.896 (0.860–0.932)
AUC-ROC	0.860	0.846	0.862	0.873 (0.834–0.912)
G.2	0.918	0.904	0.914	0.920 (0.889–0.952)
G.3	0.758	0.720	0.752	0.772 (0.722–0.821)
G.4	0.908	0.915	0.921	0.927 (0.896–0.957)
Precision				
G.2	0.617	0.576	0.560	0.583 (0.525–0.641)
G.3	0.684	0.258	0.287	0.600 (0.542–0.658)
G.4	0.914	0.911	0.928	0.930 (0.900–0.960)
Recall				
G.2	0.916	0.791	0.812	0.913 (0.880–0.946)
G.3	0.106	0.05	0.122	0.100 (0.065–0.135)
G.4	0.936	0.938	0.911	0.926 (0.895–0.957)
Specificity				
G.2	0.890	0.885	0.874	0.871 (0.831–0.910)
G.3	0.995	0.978	0.963	0.992 (0.981–1.000)
G.4	0.766	0.755	0.812	0.816 (0.770–0.861)
F1				
G.2	0.737	0.667	0.663	0.712 (0.659–0.765)
G.3	0.179	0.078	0.171	0.171 (0.127–0.216)
G.4	0.925	0.924	0.919	0.928 (0.898–0.958)

## Data Availability

The data analyzed in the current study are publicly available from: BRATS: https://www.kaggle.com/datasets/awsaf49/brats20-dataset-training-validation (accessed on 30 May 2022), EGD: https://xnat.bmia.nl/data/archive/projects/egd (accessed on 2 August 2022), REMBRANDT: https://www.nitrc.org/projects/rembrandt_brain/ (accessed on 9 September 2022).

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
