# Peer review of "AI-Based Glioma Grading for a Trustworthy Diagnosis: An Analytical Pipeline for Improved Reliability"

_cancers, 2023, doi:10.3390/cancers15133369_

Round 1
Reviewer 1 Report
Thank you very much for allowing me the opportunity to revise this interesting manuscript. AI is a promising tool in the management of patients with complex diseases.
I have some minor revisions to suggest for improving the manuscript:
- the latest WHO classification released in 2021 modify the grading of CNS tumors (1-4 instead of I-IV); the Authors need to change accordingly as also the entire description of tumor types (lines 29-43)
- the sentence: The gold-standard procedure for glioma grading involves surgery and histological 44 evaluation of the tumor. This method is invasive and time-consuming, which justifies 45 the investigation of non-invasive and pre-operative procedures to characterize the tumor, 46 accelerate the diagnosis and plan personalized treatments, is too strong. Surgery represents a unique moment not just for diagnosis but also for prognosis (there are a lot of works in which is demonstrates a fundamental role of extension of resection in OS and PFS).
Reviewer 2 Report
In this study, the authors provide an interesting non-invasive DL-based analytical pipeline for glioma grading, reliable for distinguishing lower grade from high grade gliomas but definitely improvable in accurately characterizing the precise tumor grade. Just a few suggestions to improve the paper: 1)page 1 line 35 and page 2 line 38: according to WHO 2021 classification, oligoastrocytoma does not still exists 2)page 1 line 36: “intermediate grade” is inappropriate terminology 3)please mention all possible applications of radiomics in brain tumors, for example the characterization of IDH, of MGMT, the differentiation between true progression and pseudoprogression in methylated glioblastomasMinor editing of English language required
